# Broadband Filter and Adjustable Extinction Ratio Modulator Based on Metal-Graphene Hybrid Metamaterials

**DOI:** 10.3390/nano10071359

**Published:** 2020-07-11

**Authors:** Haoying Sun, Lin Zhao, Jinsong Dai, Yaoyao Liang, Jianping Guo, Hongyun Meng, Hongzhan Liu, Qiaofeng Dai, Zhongchao Wei

**Affiliations:** Guangdong Provincial Key Laboratory of Nanophotonic Functional Materials and Devices, School of Information and Optoelectronic Science and Engineering, South China Normal University, Guangzhou 510006, China; 2018022120@m.scnu.edu.cn (H.S.); 2017021792@m.scnu.edu.cn (L.Z.); jsdai@m.scnu.edu.cn (J.D.); lyy777@m.scnu.edu.cn (Y.L.); guojp@scnu.edu.cn (J.G.); hymeng@scnu.edu.cn (H.M.); lhzscnu@163.com (H.L.)

**Keywords:** metamaterials, extinction ratio, bandwidth, EIT analogue

## Abstract

A novel multifunctional device based on a hybrid metal–graphene Electromagnetically induced transparency (EIT) metamaterial at the terahertz band is proposed. It is composed of a parallel cut wire pair (PCWP) that serves as a dark mode resonator, a vertical cut wire pair (VCWP) that serves as a bright mode resonator and a graphene ribbon that serves as a modulator. An ultra-broadband transmission window with 1.23 THz bandwidth can be obtained. The spectral extinction ratio can be tuned from 26% to 98% by changing the Fermi level of the graphene. Compared with previous work, our work has superior performance in the adjustable bandwidth of the transmission window without changing the structure of the dark and bright mode resonators, and has a high extinction ratio and dynamic adjustability. Besides, we present the specific application of the device in filters and optical modules. Therefore, we believe that such a metamaterial structure provides a new way to actively control EIT-like, which has promising applications in broadband optical filters and photoelectric intensity modulators in terahertz communications.

## 1. Introduction

Metamaterials that serve as kinds of artificial materials have received tremendous concern in modern materials science and physics [1,2] for their extraordinary material properties and flexible control of electromagnetic waves. Unlike traditional materials, metamaterials are composed of subwavelength structural units, and their dielectric constant and magnetic permeability are both negative. A series of physical concepts and the novel phenomena of metamaterials have greatly expanded people’s thinking and deepened the understanding of electromagnetics. They are widely used to realize novel optical phenomena which do not exist in natural materials, such as the negative refractive index effect [1], invisibility cloaking [3], perfect lenses [4], and electromagnetic wave perfect absorbers [5] Metamaterials have great potential in the development of new optical components [6,7], which can be realized by using the super-control ability of light. Metamaterials with adjustable functions are of great significance to this application and make it possible to develop active photonic devices, such as modulators and filters. Therefore, great efforts and progress [8,9,10,11] have been made in the research of adjustable metamaterials. As a further extension of these works, we committed to the design an actively controlled metamaterial which has an adjustable optical response in multifunctional optical devices.

Electromagnetically induced transparency (EIT) is a significant phenomenon which comes from the destructive interference between different excitation pathways in a three-level atomic system [12,13]. It can be potentially useful in many applications, such as the dynamic storage of light [14], slow-light devices [15], and nonlinear effects [16]. However, the application of EIT is extremely limited due to the fact that the generation of EIT phenomena requires harsh environmental conditions [17,18]. The appearance of metamaterials provides a way to realize classical analogs of electromagnetically induced transparency (EIT-like) [19,20,21]. In past decade, graphene with unique properties, particularly the conductivity of graphene, can be easily regulated by changing the Fermi level $E_{F}$ [22,23], and has attracted considerable attention to the active control of the EIT-like response. Our group proposed an active modulation of the amplitude of the EIT-like transmission window in terahertz-band metamaterials by changing the Fermi level $E_{F}$ of graphene [24]. A lot of similar work [25,26,27] has been done on the amplitude modulation of the transmission window. However, the transmission window of these EIT-like metamaterials has a narrow bandwidth and bandwidth adjustment range, which severely restricts the application scope of EIT-like.

In this paper, we proposed a hybrid metal–graphene metamaterial in the terahertz spectrum for the realization of changing the EIT-like transmission window and adjusting the spectral extinction ratio. After integrating two graphene strips with different widths into the metamaterial, the maximum bandwidth of the transmission window reached up to 1.23 THz by changing the graphene width, and we realized the adjustable spectral extinction ratio from 26% to 98% in the modulation process. Compared with previous works, our work has superior performance in the adjustable bandwidth range. More importantly, the realization of broadening the transmission window and adjusting the spectral extinction ratio was done by manipulating the graphene instead of changing the structure of the dark and bright mode resonators. We believe that such EIT metamaterials have promising applications in broadband slow-light devices, filters, and modulators in terahertz communications.

## 2. Materials and Methods

The schematic diagram of a hybrid metal–graphene terahertz metamaterial is shown in Figure 1a. The structural unit is comprised a parallel cut wire pair (PCWP) that serves as a dark mode resonator and a vertical cut wire pair (VCWP) that serves as a bright mode resonator. Aluminum (Al) is selected as the material for the bright and dark mode resonators. There are two graphene ribbons with different widths, one of the graphene ribbons is placed under the horizontal cut wire of the bright mode resonator for the realization of the tuning bandwidth of the transmission window, the other is the “wire” connecting all subunit structures. All structures are built on the substrate, which is composed of a dielectric with a relative permittivity of 11.7. The thickness of the substrate and the aluminum are 400 nm and 200 nm, respectively. Figure 1b describes the detailed structural parameters for a single substructure of the EIT metamaterial.

Aluminum (Al) is selected as the material for the bright and dark mode resonators, which can be described by the Drude model in the terahertz band:(1)$$\varepsilon_{Al}=\varepsilon_{\infty }-\frac{\omega_{P}^{2}}{\omega^{2}+i\omega \gamma }$$

In Formula (1), *γ* is the damping constant and $\omega_{p}$ is the plasma frequency, and their values are $1.22\times 10^{14}\;\mathrm{rad}/s$ and $2.24\times 10^{16}\;\mathrm{rad}/s$, respectively.

The conductivity of graphene is a major contributor to its properties. The conductivity of graphene is composed of the in-band and the inter-band from the Kubo equation:(2)$$\begin{array}{c} \sigma_{g}=\sigma_{\mathrm{in}}+\sigma_{\mathrm{int}er}\\ =\frac{2e^{2}K_{B}T}{\pi \hbar^{2}}\frac{i}{\omega +i\Gamma^{-1}}\ln\left[2\cosh\left(\frac{E_{F}}{2K_{B}T}\right)\right]+\\ \frac{e^{2}}{4\hbar }\left[\frac{1}{2}+\frac{1}{\pi }\tan^{-1}\left(\frac{\hbar \omega -2E_{F}}{2K_{B}T}\right)-\frac{i}{2\pi }\ln\frac{{\left(\hbar \omega +2E_{F}\right)}^{2}}{{\left(\hbar \omega -2E_{F}\right)}^{2}+4{\left(K_{B}T\right)}^{2}}\right] \end{array}$$
where $K_{B}$ is the Boltzmann constant, e is the electron charge, *T* is environment temperature which is set to 300 *K*, ω is the incident plane wave angle frequency, and *ħ* is the simplified Planck constant. It is worth noting that the Fermi level of graphene is represented by $E_{F}$ in the formula, and the average relaxation time is represented by *Γ*.

In the terahertz region, the in-band partial conductivity can be neglected for the Pauli exclusion principle. Under this condition: *E_F_ ≫*
*K_B_T and E_F_ ≫*
*ħω*, and $\sigma_{g}$ can be expressed by a simplified Drude formula [28].
(3)$$\sigma_{g}={\left(\frac{e}{\hbar }\right)}^{2}\frac{E_{F}}{\pi }\frac{i}{\omega +i\Gamma^{-1}}$$

The carrier relaxation time is $\;\Gamma =(\mu E_{F})/(e)$, in which $V_{F}=1.1\times 10^{6}\;m/s$ represents the Fermi velocity, and $\mu {=3000\;\mathrm{cm}}^{2}/(V\times s)$ is the carrier mobility [29]. Figure 2 depicts the conductivity of graphene under different Fermi levels.

## 3. Results

To research the optical response of the proposed EIT-like metamaterials, finite-difference time-domain (FDTD) software is used in our work. In the simulation calculation, the basic settings are as follows: the background index is set as 1.0, the environment temperature is 300 T, periodic boundary conditions are set in the x- and y-directions, and there are perfect matching layers in the z-direction. Figure 3a shows the transmission spectra of the PCWP, VCWP, and the entire structure. It can be seen from this figure that the spectral characteristic of the VCWP indicates that there is a symmetric Lorenz-type resonance, where its central frequency is located at the position of 2.14 THz. By comparison, the PCWP exhibits no optical response to incident light and almost completely transmits in the terahertz band. An EIT-like transmission window with a bandwidth of 0.33 THz can be achieved under the condition of the combination of the VCWP and PCWP, when there is polarization illumination of the plane wave along the *x*-direction.

In order to reveal the physical mechanism of the transmission window in the proposed EIT metamaterials, we plot the transmitted electric field distributions of the individual VCWP and PCWP and the combination metamaterials at a 2.14 THz resonant frequency, as shown in Figure 3b,d. Figure 3b shows that the VCWP, which exhibits strong coupling with incident plane waves, can excite a partial surface plasmon (PSP) [30]. Figure 3c is an electric field distribution graph of the PCWP, showing that it is not excited at the corresponding frequency, which has a very weak electric field distribution due to the structural symmetry concerning the polarization of the incident plane wave. In this sense, we can say that the VCWP and the PCWP are the bright and dark mode resonators, respectively. Caused by the bright mode via near field coupling, the dark mode can be excited by the local field. Figure 3d depicts the electric field distribution of the EIT metamaterial at 2.14 THz. When the PCWP and VCWP are integrated into the unit cell, it is obvious that there are strong electric fields concentrated at two corners of the PCWP, which indicates that the PCWP switches to monopole mode after excitation by the VCWP. The destructive interference between the dipole mode and monopole mode gives rise to the generation of EIT-like metamaterials.

### 3.1. Graphene-Based Tunable Metamaterial Broadband Filter

The local schematic diagram of the structure is shown in Figure 4a. Figure 4b shows the simulated transmission spectrum of the proposed structure with different graphene widths g (when the Fermi level is fixed at 1.2 eV). The length of g increases from 0.8 to 4 µm with a step of 0.8 when the widths of both ends of the horizontal graphene are changed. It can be seen from Figure 4c that when the graphene width *g* is less than 2.4 µm the bandwidth increases linearly and the left transmission dip shifts gradually towards the blue spectrum, but the right transmission dip is almost unchanged. Therefore, the bandwidth of the transmission window increases with the width of the graphene. For visual observation, we plotted the transmission window with various lengths g in Figure 4c. The bandwidth of the transmission window can be adjusted from 0.4 to 1.23 THz. Graphene acts as a special energy bridge, which results in the change in resonance intensity at the same frequency of bright and dark mode, and consequently changes in the transmission window bandwidth. It is noteworthy that the bandwidth is adjusted only by graphene parameters rather than by changing the structure of the bright and dark modes resonators.

The designed structure has superior performance in the adjustable bandwidth range, which has promising applications in broadband filters. We made the application diagram of the device as shown in Figure 5. Using the frequency adjustable characteristic of the device, the spectrum width can be adjusted when the signal passes through the sample. Devices can be applied to filters. We compared the performance of our devices with that of the current articles, which is summarized in Table 1.

### 3.2. Graphene-Based Tunable Metamaterial Electro-Optic Modulator

Next, we show the transmission window of the condition of the graphene at different Fermi levels. The DC voltage source is connected to the gold bar of the metamaterial which is linked with the graphene. Therefore, the characteristics of graphene can be changed by altering the DC voltage, so the active control of EIT can be realized. Figure 6 shows the transmission spectra with Fermi levels from 0.4 to 1.2 eV. After adding graphene with a Fermi level of 0.4 eV, compared with the absence of graphene, the left transmission dip with an amplitude of 0.64 shifted to 0.71 THz and the resonance frequency of the right transmission dip is almost unchanged. As the Fermi level E_F_ increased from 0.4 eV to 1.2 eV, the amplitude of the left and right transmission dips gradually declines. At the Fermi level of 1.2 eV, the maximum modulation depths ($MD=\left|T_{Max}-T_{Min}\right|/T_{Max}$) of the left and right dips are 85% and 92%, respectively. We employ the spectral extinction ratio to evaluate the performance of the EIT metamaterial devices, and it is described as [37]:(4)$$S_{\mathrm{con}}=\frac{\left(T_{peak}-T_{dip}\right)}{\left(T_{peak}+T_{dip}\right)}\times 100\%$$
where $T_{peak}$ is the intensity of the transmission peak and $T_{dip}$ is the intensity of the transmission dip. In the modulation process, $S_{\mathrm{con}}$ can be dynamically tunable from 26% to 98% by increasing the Fermi level of the graphene. Therefore, the proposed EIT metamaterial has promising applications in optical filters and modulators.

To explain the origin behind this phenomenon, we plot the corresponding transmitted electric field |E| distributions of graphene at 0.4 eV and 1.2 eV at a 0.82 THz resonant frequency.

As we can see in Figure 7a, when the Fermi level of graphene is at 0.4 eV, it hinders the energy transfer of the electric current between the VCWP resonator and the PCWP resonator for its low conductivity, which inhibits the formation of EIT and thus leads to the transmission dips with high amplitude. As the Fermi level of graphene reaches up to 1.2 eV, the conductivity of the graphene also increases (according to Figure 2), which makes the energy of the VCWP more easily transferred to the PCWP through the graphene, thus the change in the electric field is more pronounced. The electric field intensity in the VCWP significantly declines while that in the PCWP obviously increases. Here, almost all the energy in the bright mode resonator is transferred to the dark mode resonator and high intensity electric fields concentrate at both ends of the dark mode resonator. By adjusting the Fermi level, the active modulation of the spectral extinction ratio can be achieved.

The extinction ratio is a significant parameter to measure the performance of optical modules. The larger the extinction ratio is, the stronger the resolution of the optical signal is, so the sensitivity is improved. However, if the extinction ratio is too large, the optical signal transmission will be easily distorted and there will be an error code at the terminal of the communication link. Therefore, it is necessary to dynamically adjust the extinction ratio according to the actual application requirements. Figure 8 shows the specific application scenario. The sample can adjust the extinction ratio under the control of voltage. We can adjust the extinction ratio by voltage by using the device in combination with the optical path, as shown in the figure. Such devices can be used in adaptive applications. When the extinction ratio is too large, the power sensor gives the device a feedback voltage to reduce the light passing rate. On the contrary, when the extinction ratio to the receiver is small, the light passing rate of the device can be increased through the feedback voltage. This kind of power control through voltage has a very important application in communication. We compare the performance of our devices with a traditional device and summarize it in Table 2. The response speed of our device is much faster than that of traditional devices due to voltage regulation. Our structure is not sensitive to temperature, which reduces the influence of temperature. In addition, our devices have good performance in an adjustable range.

## 4. Prospects for Future Work

Our group proposed an active modulation of the amplitude of the EIT-like transmission window by changing the Fermi level $E_{F}$ of graphene. This work proves that the bandwidth and extinction ratio of EIT can be tuned by graphene without changing the structure of bright and dark mode resonators. In the future, we will attempt to design a kind of metamaterial which can modulate the amplitude, extinction ratio, and bandwidth of EIT at the same time only by changing the Fermi level $E_{F}$ of graphene, which will achieve complete EIT control without changing any structural parameters.

## 5. Conclusions

In summary, we propose a tunable transmission window bandwidth and spectral extinction ratio EIT-like metamaterial based on hybrid metal–graphene metamaterials in the terahertz band. It is composed of a parallel cut wire pair (PCWP) resonator, a vertical cut wire pair (VCWP) resonator, and two graphene ribbons with different widths. We plot the electrical conductivity of graphene at different Fermi levels. Through the simulation calculation, it is verified that an EIT-like transmission window with a bandwidth of 0.33 THz can be obtained through the magnetic dipole coupling between bright–dark modes. The width of the EIT transmission window can be changed by altering the lengths of both ends of horizontal graphene. The transmission window bandwidth modulation is from 0.4 to 1.23 THz. By controlling the gate voltage to change the Fermi level of graphene, the spectral extinction ratio can be adjusted from 26% to 98% and we then analyze the physical mechanism behind it, which shows that graphene with different Fermi energy levels can control the electric field energy of the device. Based on the adjustable function of the bandwidth and extinction ratio, we give a specific application and compare it with previous work. The results show that the performance and method of the device are improved. Finally, we summarize the inspiration of this work and the direction of the next work. In a way, such an EIT-like metamaterial has promising applications in broadband filters and modulators in terahertz communications.

## Figures and Tables

**Figure 1 nanomaterials-10-01359-f001:**
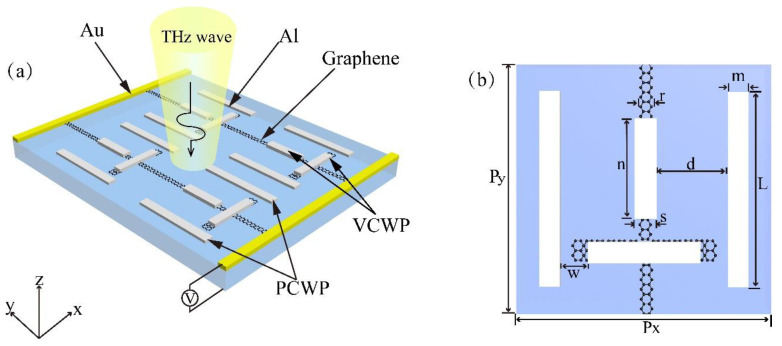
(**a**) The schematic of a hybrid metal–graphene metamaterial under a plane wave incident in the *x*-polarization direction; (**b**) detailed structural parameters of metamaterials: *P_x_* = 80 µm, *P_y_* = 120 µm, *L* = 60 µm, *m* = 5 µm, *w* = 4.5 µm, *n* = 42 µm, *s* = 4 µm, *r* = 1 µm, *d* = 23.5 µm perpendicularly.

**Figure 2 nanomaterials-10-01359-f002:**
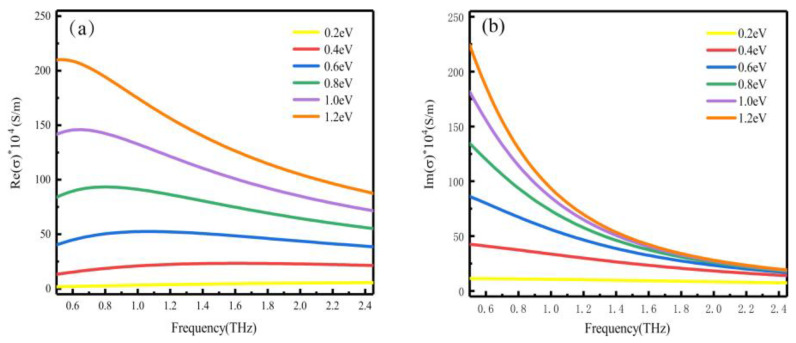
(**a**) The real and (**b**) the imaginary parts of the graphene conductivity with the Fermi level changing from 0.2 eV to 1.2 eV.

**Figure 3 nanomaterials-10-01359-f003:**
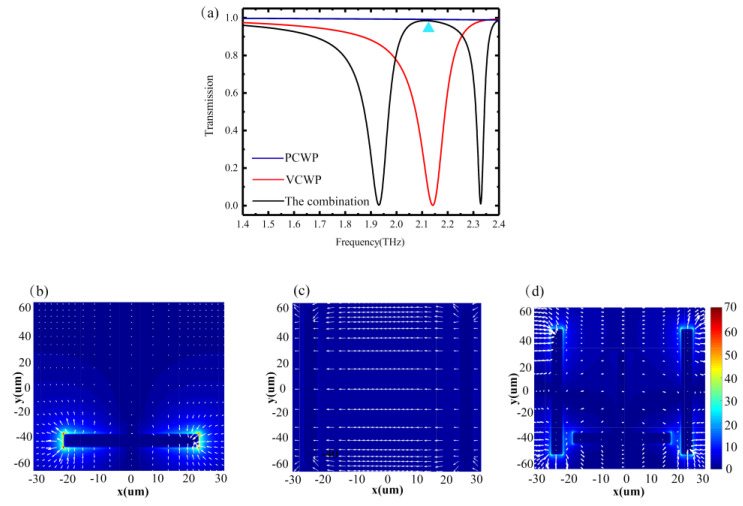
(**a**) Transmission spectra of the parallel cut wire pair (PCWP) alone, the vertical cut wire pair (VCWP) alone, and the EIT metamaterial. Distribution of the local electric field of (the blue arrow points to 2.14 THz) (**b**) the VCWP, (**c**) the PCWP, and (**d**) the combined EIT structure at a frequency of 2.14 THz.

**Figure 4 nanomaterials-10-01359-f004:**
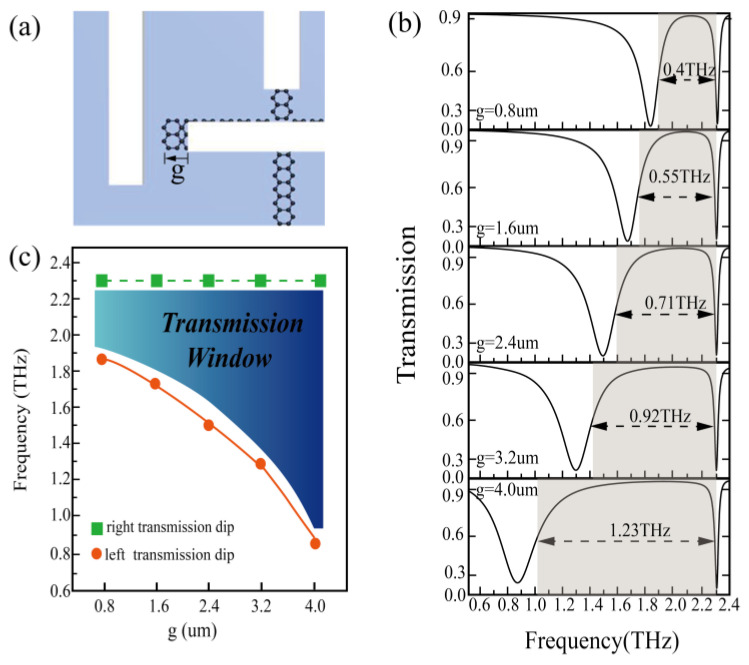
(**a**) Partial schematic of the proposed EIT-like metamaterial, (**b**) resonance frequency shift under different *g*, (**c**) transmission window under different *g*.

**Figure 5 nanomaterials-10-01359-f005:**
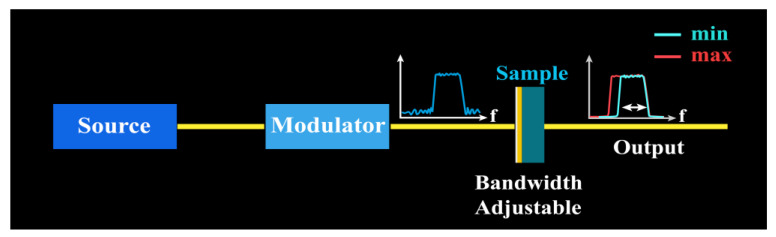
Application of an adjustable filter.

**Figure 6 nanomaterials-10-01359-f006:**
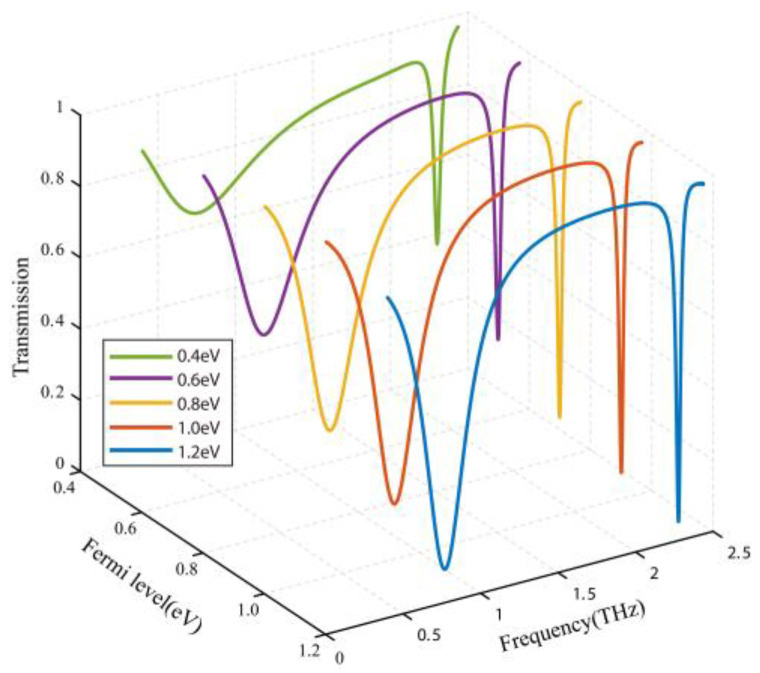
Spectrum of the proposed EIT metamaterial at different Fermi levels of graphene.

**Figure 7 nanomaterials-10-01359-f007:**
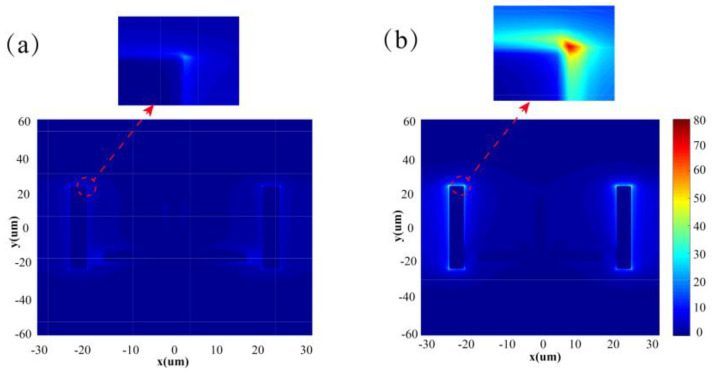
The electric field distribution of the metamaterial on the surface at a frequency of 0.82 THz when the graphene Fermi level is 0.4 eV (**a**) and 1.2 eV (**b**), respectively.

**Figure 8 nanomaterials-10-01359-f008:**
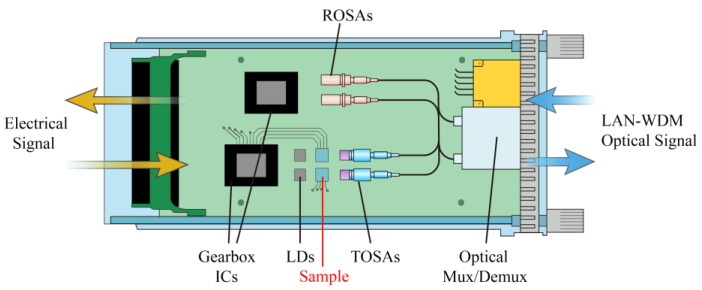
Application of extinction ratio regulation of devices in an optical module.

**Table 1 nanomaterials-10-01359-t001:** The performance comparison of different EIT-like metamaterial structures.

Structure	Bandwidth Tunable Range	Adjustment Method	Reference and Year
two U-shaped split rings	0.16–0.28 THz	By changing dark mode	[31] 2015
a cut wire ring, a couple of U-shaped split ring	0.03–0.10 THz	By changing dark mode	[32] 2017
a cut wire ring, a pair of split rings	0.42–0.78 THz	By changing dark mode	[33] 2017
a cut wire, two H-shaped split rings	1.05–1.64 THz	By changing dark mode	[34] 2018
a U-shaped (USR), a L-shaped split ring	0.34–0.42 THz	By changing dark mode	[35] 2018
a CWR, a pair of U-shaped split rings	0.15–0.41 THz	By changing dark mode	[36] 2018
a PCWP, a VCWP (Our work)	0.4–1.23 THz	By changing graphene	

**Table 2 nanomaterials-10-01359-t002:** Comparison between our device and a traditional device.

Related Parameters	Traditional Device	Sample Proposed Device
Modulation method	Temperature compensation circuit	Electric regulation
Response time	Millisecond level	Picosecond level
Influence of temperature on extinction ratio	Nonlinear increase (228–353 T)	Not changed (228–353 T)
Adjustable range	10.3–14.5 dB	3.4–18.5 dB

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
