# Peer review of "Broadband Filter and Adjustable Extinction Ratio Modulator Based on Metal-Graphene Hybrid Metamaterials"

_nanomaterials, 2020, doi:10.3390/nano10071359_

Round 1

Reviewer 1 Report

The authors report a novel terahertz resonator structure based on a hybrid metal-graphene EIT. The work is clearly an extension of their previous studies but has improved bandwidth and a tuneable spectral extinction ratio. They are able to present an application scheme in terms of an adjustable wideband filter, with properties potentially superior to other results in the literature.

There would seem some argument as to the extent of the advance on previous work (Nanomaterials, 2019, 9(2): 171) I would tend to agree that the adjustment of the transmission window bandwidth and the optimisation of the  extinction has not been demonstrated in the  previous work and that therefore there is sufficient novelty in this paper. The table they have produced does a good job in illustrating the differences.

The English grammar is a little bit marginal in places and ideally needs to be improved somewhat.  At times it affects the understanding of the results and their discussion. I also found the introduction a bit repetitive and laboured in places and then, in contrast, the discussion and conclusions are a bit short. Some really important aspects such as control of the graphene parameters get only a very short comment.

Tables like the one with a comparison of different EIT-like metamaterial structures (note there is no table number) and table 2 need a written discussion. Otherwise it is an over-simplistic comparison.

Reviewer 2 Report

This manuscript entitled “Broadband Filter and Adjustable Extinction Ratio Modulator Based on Metal-graphene Hybrid Metamaterials,” by Haoying Sun et al., reports tunable terahertz transmission window bandwidth and spectral extinction ratio based on metal-graphene metamaterials. The authors proposed a metamaterial composed of a parallel cut wire pair (PCWP), a vertical cut wire pair (VCWP) resonator and two graphene ribbons by the numerical simulations. The suggested EIT-like metamaterial demonstrated transmission window bandwidth modulation from 0.4 to 1.23 THz by controlling the Fermi level of graphene. They also proposed promising applications on broadband filters and modulators in the terahertz communications. However I do not recommend publication of this work in nanomaterials for the lack of development.

This report should show a result of fabricated hybrid metal-graphene metamaterials experimentally and compare with the results from numerical simulation. In this paper, they compare the performance of different EIT-like metamaterials [1], [2], [3], [4]. These previous works fabricated and measured the metamaterial structures and compare with simulation. I wonder whether the results of the experiment can be achieved with proposed performance in this report.

In previous report [5], they also depicts the conductivity of graphene under different Femi levels. Compare this with Figure 2, it is different result even they depicted from same formula. Also, the scale of conductivity seems too high (~106) in Figure 2 (b)

[1] Su, Xiaoqiang, et al. “Broadband Terahertz Transparency in a Switchable Metasurface.” IEEE Photonics Journal, vol. 7, no. 1, IEEE, 2015, pp. 1–8, doi:10.1109/JPHOT.2015.2390146.

[2] Yahiaoui, R., et al. “Electromagnetically Induced Transparency Control in Terahertz Metasurfaces Based on Bright-Bright Mode Coupling.” Physical Review B, vol. 97, no. 15, American Physical Society, 2018, pp. 1–5, doi:10.1103/PhysRevB.97.155403.

[3] Yahiaoui, Riad, et al. “Active Control and Switching of Broadband Electromagnetically Induced Transparency in Symmetric Metadevices.” Applied Physics Letters, vol. 111, no. 2, 2017, doi:10.1063/1.4993428.

[4] Zheng, Xiaobo, et al. “Broadband Terahertz Plasmon-Induced Transparency via Asymmetric Coupling inside Meta-Molecules.” Optical Materials Express, vol. 7, no. 3, 2017, p. 1035, doi:10.1364/ome.7.001035.

[5] Lao, Chaode, et al. “Dynamically Tunable Resonant Strength in Electromagnetically Induced Transparency (EIT) Analogue by Hybrid Metal-Graphene Metamaterials.” Nanomaterials, vol. 9, no. 2, 2019, doi:10.3390/nano9020171.
